# Exercise Capacity and Physical Activity in Non-Cystic Fibrosis Bronchiectasis after a Pulmonary Rehabilitation Home-Based Programme: A Randomised Controlled Trial

**DOI:** 10.3390/ijerph191711039

**Published:** 2022-09-03

**Authors:** Sindy Cedeño de Jesús, Virginia Almadana Pacheco, Agustín Valido Morales, Ana Miriam Muñíz Rodríguez, Rut Ayerbe García, Aurelio Arnedillo-Muñoz

**Affiliations:** 1Respiratory Department, Virgen Macarena University Hospital, 41009 Seville, Spain; 2Physical Medicine and Rehabilitation, Virgen Macarena University Hospital, 41009 Seville, Spain; 3Respiratory, Allergology and Thoracic Surgery Department, Puerta del Mar University Hospital, 11009 Cadiz, Spain

**Keywords:** non-cystic fibrosis bronchiectasis, pulmonary rehabilitation, exercise capacity, physical activity, cardiopulmonary exercise test

## Abstract

Background: Patients with chronic respiratory disease have low exercise capacity and limited physical activity (PA), which is associated with worsening dyspnoea, exacerbations, and quality of life. The literature regarding patients with non-cystic fibrosis bronchiectasis (non-CF BQ) is scarce, especially regarding the use of cardiopulmonary exercise tests (CPET) to assess the effects of home-based pulmonary rehabilitation programmes (HPRP). The aim was to evaluate the effect of an HPRP on the exercise capacity of non-CF BQ patients using CPET and PA using an accelerometer. Methods: Our study describes a non-pharmacological clinical trial in non-CF BQ patients at the Virgen Macarena University Hospital (Seville, Spain). The patients were randomised into two groups: a control group (CG), which received general advice on PA and educational measures, and the intervention group (IG), which received a specific 8-week HPRP with two hospital sessions. The variables included were those collected in the CPET, the accelerometer, and others such as a 6 min walking test (6MWT) and dyspnoea. The data were collected at baseline and at an 8-week follow-up. Results: After the intervention, there was a significant increase in peak VO_2_ in the IG, which was not evidenced in the GC (IG 66.8 ± 15.5 mL/min *p* = 0.001 vs. CG 62.2 ± 14.14 mL/min, *p* = 0.30). As well, dyspnoea according to the mMRC (modified Medical Research Council), improved significantly in IG (2.19 ± 0.57 to 1.72 ± 0.05, *p* = 0.047) vs. CG (2.07 ± 0.7 to 2.13 ± 0.64, *p* = 0.36). In addition, differences between the groups in walked distance (IG 451.19 ± 67.99 m, *p* = 0.001 vs. CG 433.13 ± 75.88 m, *p* = 0.981) and in physical activity (IG 6591 ± 3482 steps, *p* = 0.007 vs. CG 4824 ± 3113 steps, *p* = 0.943) were found. Conclusion: Participation in a specific HPRP improves exercise capacity, dyspnoea, walked distance, and PA in non-CF BQ patients.

## 1. Introduction

Non-cystic fibrosis bronchiectasis (non-CF BQ) is defined as a chronic inflammatory bronchial disease with irreversible dilation of the bronchial lumen, which can be brought on by several different causes [1]. Patients commonly present with chronic bronchial infection leading to a progressive decline in lung function [2], which can result in a lowering of their exercise capacity and physical activity (PA) [3]. These limitations could be due to ventilatory limitation or physical deconditioning, although it is not clear whether these limitations appear in the disease or are linked to more advanced and severe stages of the disease [4].

Nowadays, pulmonary rehabilitation programmes (PRP) are one of the main objectives and pillars of the non-pharmacological therapeutic strategy for chronic respiratory patients. They provide a multidisciplinary intervention, which includes physical preparation and educational and physiotherapy measures.

In this way, PRP designed to promote PA have shown to improve dyspnoea, fatigue, exercise tolerance, and quality of life after specific training [5]. In general, other benefits of exercise include delayed loss of lung function [6], improved oxygen consumption, muscle strengthening, physical benefits, reduction in hospitalisations and, by extension, morbidity reduction [7].

Beneficial PRP have been described to include treadmill or ground walking exercises with an initial intensity of 75–80% of the maximal speed achieved in the incremental walking test (ISWT) or 80% of the maximal oxygen consumption (VO_2_max) in the cardiopulmonary exercise test (CPET), as well as upper and lower limb strength training with free weights and/or body weight or symptom-limited training [8,9].

Some studies in non-CF BQ after a home pulmonary rehabilitation programme (HPRP) have reported increased peripheral muscle strength [10], time to first exacerbation [11], improved exercise tolerance [10,12,13], and increased walked distance [14,15]. Nevertheless, these changes that occur in exercise capacity and PA level are heterogeneous and have low-certainty evidence [7].

On the other hand, it is known that not all patients included in PRP have the same response to exercise, and these data are routinely investigated and published in chronic obstructive pulmonary disease (COPD). It is reported that 20–40% of patients do not respond to PRP, but this is less well known in non-CF BQ [16].

However, one systematic review [7] has shown that supervised exercise or a PRP together with professional advice for at least 8 weeks improves PA [17], walked distance, and health-related quality of life [8].

Despite the evidence on the benefits of PR, few programmes are available for non-CF BQ patients due to barriers in accessibility or assistance and programme design, which is where HPRPs can provide a useful alternative. Arguing in their favour, a recent study based on a home-based programme concluded that clear benefits could be obtained in exercise capacity, dyspnoea, and walked distance [10] just by changing the patients’ level of PA [13,14]. 

In general, submaximal exercise tests, including 6-min walk test (6MWT) and ISWT, have become the standard way to evaluate the effects of antibiotic treatments, airway clearance techniques, or inspiratory muscle training [13] in patients with non-CF BQ. However, very few studies have assessed the benefits of a pulmonary rehabilitation (PR) programme [10,18] with maximal exercise tests [9,13]. In this context, the CPET for oxygen consumption (VO_2_) is gaining ground in clinical settings as it provides information on the patient’s general physical state, limitations during exercise, and changes after the therapeutic interventions have been performed. 

Against this background, the purpose of this study was to evaluate the effect of a pilot home-based pulmonary rehabilitation programme (HPRP) on maximum exercise capacity using CPET and PA using an accelerometer in non-CF BQ patients.

## 2. Materials and Methods

### 2.1. Design 

This open randomised clinical trial was conducted with patients from a non-CF BQ. The participants were prospectively recruited from a specialised bronchiectasis consultation at the Virgen Macarena University Hospital (Seville, Spain) from June 2018 to January 2020. The participants were assessed at baseline (pre-intervention, HPRP) and immediately post-intervention, with an 8-week follow-up. The recruitment and data collection were conducted identically for both groups and at the same locations. 

The trial protocol was registered at www.clinicaltrials.gov (accessed on 18 May 2022) (NCT05369624). The study followed the CONSORT recommendations for non-pharmacological clinical trials [19]. 

The project complied with the ethical principles of the Declaration of Helsinki and current Spanish legislation (RD 1090/20215). A signed informed consent form was provided prior to inclusion in the study. The data evaluated were obtained under strict confidentiality rules, following the Organic Law on Personal Data Protection guaranteeing digital rights 3/2018, dated 5 December. The study was approved by the ethics and clinical research committee of the Virgen Macarena-Virgen del Rocio University Hospitals.

### 2.2. Participants

The inclusion criteria were a certain diagnosis of non-CF bronchiectasis (chest high-resolution computed tomography), aged over 18 years old, clinically stable in the previous 6 weeks (no need for antibiotic therapy due to exacerbations), a dyspnoea grade of 1 according to the modified Medical Research Council (mMRC) dyspnoea scale, and the ability to perform the tests and the training protocol. The exclusion criteria were the coexistence of COPD or being a former or active smoker with more than 10 pack–years.

The participants were consecutively assigned using computer-generated numerical sequence randomisation into two groups: A control group (GC) and an intervention group (IG), consisting of a home-based respiratory rehabilitation programme. 

#### 2.2.1. Intervention 

The groups were homogeneous in terms of baseline characteristics as well as age and sex. The intervention carried out was developed as follows:

CG: The participants received written instructions on how to perform PA and progressively increased walking intensity. It was recommended to perform it at least between 3–5 times a week (to facilitate compliance for at least 3 days) for 30 min. This group did not receive supervision during the study. 

IG: The intervention group of patients conducted an HPRP, which included educational measures and the learning of self-management techniques and the drainage of secretions. This group received two hospital sessions at the respiratory rehabilitation gymnasium. The visit was individualised and supervised by a physiotherapist: in the first session, an educational workshop was held, explaining different aspects of the disease, inhaler technique, dietary recommendations, and self-management of the disease. In addition, the exercises to be performed at home were explained, indicating a frequency of 3 to 5 times per week. The second session included 1 h of exercise and 45–60 min of education. Reminder and motivational calls were carried out weekly for 8 weeks. Control of the training, form, duration, intensity, and increase in the exercise according to the symptoms (maximum heart rate, dyspnoea, and fatigue according to the modified Borg scale <4) were established as markers of training intensity. The patients were given a diary for daily PA collection, including the frequency and duration of the same. We considered good adherence for patients who fulfilled at least 70% of the recommended activity and who completed the programme.

#### 2.2.2. Strength Training

Included upper and lower limb exercises, initially with no weights but weights were progressively once a week depending on symptoms, in 2 sets with 6–8 repetitions, for 3–5 days a week. The exercises recommended were ‘hanger’ (that exercise the latissimus dorsi muscle), ‘butterfly’ (pectoralis major muscle), ‘neck press’ (triceps brachii and deltoids), ‘leg flexion’ (biceps femoris and gastrocnemius), and ‘leg extension’ (quadriceps femoris).

#### 2.2.3. Endurance Training 

The patients could choose between walking or cycling 3–5 days per week, for at least 20 min, increasing the duration of the exercise weekly, depending on their symptoms.

#### 2.2.4. Measurement of Variables

The assessments included demographic, clinical, and microbiological isolates that were collected from their medical records.

The measurements of pulmonary function were taken using a Jaeger Viasys Mastercope spirometer^®^ (Hoechberg, Germany), using measurements of forced expiratory volume in 1 s (FEV1 mL, FEV1%), forced vital capacity (FVC mL and FVC%), and the FEV1/FVC ratio. 

To evaluate exercise capacity, we used a Jaeger-CareFusion Vyntus CPX ergospirometer^®^ (Hoechberg, Germany). The participants underwent a symptom-limited incremental exercise test on an ergometric bicycle. The test was performed following a stepped incremental protocol at a rate of 10–15 w/min [20], maintaining a constant speed of 60 r.p.m. The heart rate (HR) was monitored using a 12-lead ECG, and the peripheral arterial oxygen saturation (SpO_2_) was measured using pulse oximetry. The expired gases were analysed, prioritising peak VO_2_, carbon dioxide production (VCO_2_), minute ventilation (VE), tidal volume (VT), and pulse oxygen (VO_2_/HR). The breathing reserve (BR) was calculated as BR (predicted maximum minute ventilation estimated from MMV = FEV1 × 35). HR peak was expressed as a percentage of the maximum predicted HR, calculated as HR max = 210 − (0.65 × age). The predicted values for VO_2_ max were calculated from the reference equations [21,22]. After exercise, the participants were asked to score their sense of breathlessness and muscle fatigue at peak exercise using the Borg scales. We considered 2.5 mL/kg/min as the smallest VO_2_ effect that would be important to detect, stating that any smaller effect would not be of clinical or substantive importance [23].

An additional assessment was carried out using the 6MWT [24]. It was conducted following standardised guidelines. A minimum of two tests were performed as a baseline assessment, and the best distance was recorded. The variables collected were walked distance (m), pre- and post-6MWT peripheral oxygen saturation (SpO_2_), heart rate at rest before exercise and immediately stopped (bpm), Borg scale, and lower limb fatigue [25]. To assess the changes from the baseline, a minimal clinically important difference (MCID) of 25 m was considered [26].

The PA level was assessed using a Multisensor ArmBand accelerometer (SenseWear mini armband; BodyMedia, Pittsburgh, PA, USA) [27] positioned on the right triceps, which recorded activity for 5 days (used from Wednesday to Sunday on all participants before and after the HPRP. It was used at home, and they were instructed to leave it on and only take it off when showering). The data were analysed using the Innerview professional version 8 software, BodyMedia, Pittsburgh, PA, USA. The participants were classified as follows: sedentary, <5000 steps/d; low active, 5000–7499 steps/day; somewhat active, 7500–9999 steps/day; active, 10,000–12,499 steps/day; and highly active, >12,500 steps/day [28].

The severity and prognosis of non-CF BQ were measured using the E-FACED scale (FEV1%, age, colonisation by Pseudomonas aeruginosa, dyspnoea according to mMRC, and exacerbations in the last year) [29].

### 2.3. Statistical Analysis

Sample size: An effect size of 0.6 was expected based on an α-error of 0.05 and a β-error of 0.20, providing sufficient power to this outcome. The standardised effect size was calculated using Cohen’s D (0.2 represents a small effect size, 0.4 represents a medium effect size, and a value of 0.6 represents a large effect size). Therefore, a minimum of 26 participants were considered to evaluate the changes in peak VO_2_ after HPRP. However, 20% was added due to possible non-completion, missing data, and dropouts [30].

The statistical analysis was performed using the statistical analysis software IBM SPSS Statistics (SPSS v23, Chicago, IL, USA). For the quantitative variables, the values were expressed as mean ± standard deviation. The qualitative variables were expressed as frequencies and percentages. The Shapiro–Wilk test was used to determine the normality of the data for all of the variables. The Student’s t-test was used to analyse the baseline differences between the groups if the data behaved according to a normal distribution. Otherwise, the Mann–Whitney U test for independent samples was used. The intra-group comparisons were carried out with the T-Student test for related samples when the data followed a normal distribution, while a Wilcoxon test was used for data with abnormal distributions. In all cases, the minimum level of significance was *p* < 0.05. 

## 3. Results

A total of 34 patients with stable non-CF BQ were randomised into two groups: 18 patients were assigned to the IG, and 16 patients were assigned to CG. Two of the IG participants dropped out of the programme after the initial assessment due to surgical intervention (*n* = 1) and a refusal to participate (*n* = 1). One CG patient refused to continue at the 5^th^ week of follow-up. Thirty-one patients were finally analysed (Figure 1). No incidences or adverse effects of the exercise programme were reported. There were no differences in the baseline characteristics of the groups (Table 1). Overall, 50.2% of the participants in the IG confirmed that they had performed the recommended training programme, compared with only 13.4% in the CG. 

The sample consisted mostly of women (CG 66%, IG 81.2%), the mean age in the CG was 59.43 ± 93, and in the IG, it was 63 ± 6.1, with overweight (Table 1). The main aetiology of the non-CF BQ in the total sample was postinfectious (35.5%), followed by idiopathic (16.1%). The distribution of the CG and IG can be found in Table 1. The mean score on the E-FACED scale was mild in both groups (see distribution by severity in Table 1), and chronic colonisation by *Pseudomonas aeruginosa* was present in 33.3% of the CG and 37.5% of the IG. In the pulmonary function tests, the CG presented with a fixed airflow obstruction, which was not confirmed in the IG, and no statistically significant differences were found (Table 1).

### 3.1. Dyspnoea

After the intervention an improvement in dyspnoea according to the mMRC (modified Medical Research Council) scale was recorded in the IG compared to the CG: baseline dyspnoea improved from 2.19 ± 0.57 to 1.72 ± 0.05 (*p* = 0.047, effect size 0.66) in the IG compared with 2.07 ± 0.70 to 2.13 ± 0.64 (*p* = 0.06, effect size 0.09) in the CG (Table 2). 

### 3.2. Six-Minute Walk Test 

In the same way, in the 6MWT, the walked distance improved more in the IG than in the CG: the baseline 6MWT improved from 401 ± 84.6 m to 451.19 ± 66.27 (*p* = 0.006, effect size 0.92) in the IG compared with 398 ± 70.6 m to 433.1 ± 75.8 m (*p* = 0.27, effect size 0.27) in the CG (Table 2). 

Although the differences were statistically significant and with minimal clinically important differences (MCID) in the IG (mean difference of 50 m, 95% IC −98.99–26.72 m), it should be noted that the CG had an increase too (mean difference of 35 m, 95% IC −97.63–−24.11), which is clinically relevant (Figure 2).

Considering the clinically relevant changes in the walked distance (greater than 25 m) in relation to their baseline, this occurred in the IG in twelve (75%) participants and in CG in seven (46.6%) (Figure 2). There were no statistical differences in the variables of peripheral oxygen saturation, heart rate, or Borg scale. 

### 3.3. Cardiopulmonary Exercise Test 

In the CPET, all of the participants stopped the test when limited by symptoms. At baseline, a low aerobic capacity (VO_2_ peak) was observed in both groups, although it should be mentioned that the RER was under 1.1 in most cases (Table 3). However, VO_2_ showed an improvement in the IG after intervention (62.9 ± 15.8 to 66.8 ± 15, *p* = 0.01, effect side 0.83), which was not evidenced in the CG (59.8 ± 14.6 to 62.2 ± 14.14, *p* = 0.30, effect size 0.32) (Table 3). No changes were detected in the other variables except for the final heart rate (HR), which decreased in the CG after 8 weeks and cannot be considered a training effect. In addition, it is worth noting that twelve (75%) of the IG participants were responders, while only eight (53%) were from the CG (Figure 2). 

Comparing the changes before and after the intervention, no significant differences or correlations were observed in exercise capacity in relation to the severity level (E-FACED) of non-CF BQ. Similarly, there was no correlation between VO_2_ and lung function.

In relation to PA, the regular daily PA was low in both groups (with activities with low intensity and low energy consumption). Although, there was an increase in the mean number of steps per day in the IG (IG from 4578 ± 3424 to 6591 ± 3482 steps (*p* = 0.007); and the CG from 4793 ± 3236 to 4824 ± 3113 steps (*p* = 0.94)) (Table 3). The PA, according to the number of steps, was distributed in: <4999 steps: CG 3 and IG 2; between 5000 and 7499 steps: CG 8 and IG 6; between 7500 and 9999 steps: CG 2 and IG 4 and over 10,000 steps: CG 2 and in the IG 4. Secondly, it was found that the IG patients increased their activity in terms of percentage compared to the CG, moving to moderate levels from 25% to 43.8%. Regarding the METs and hours of PA evaluated before and after, no notable changes were found in either group. 

No significant differences or correlations were observed in exercise capacity, VO_2_ peak, lung function, or severity level (E-FACED) with PA.

## 4. Discussion

In our study, we evaluated the impact of an RPHP in non-CF BQ patients. The most relevant finding was that the patients who carried out the supervised respiratory home-based rehabilitation programme showed significant improvements in VO [28], PA (number of steps and intensity of activity), walked distance, and dyspnoea, compared to those who only received general recommendations. Our results support the fact that home-based rehabilitation programmes are valid and accessible for incorporation into daily clinical practice without great cost. 

Moreover, one of the most novel findings of our study is the confirmation that non-CF BQ patients showed low exercise capacity [7], reflected in the low peak VO_2_, which can possibly be improved after a specific home-based training programme. To our knowledge, this is one of the first studies that has evaluated exercise capacity with CPET after HPRP exclusively in non-CF BQ patients and its association with severity.

The use of VO_2_ max after training is the classic way to evaluate training efficiency [31]. Although there is no baseline value for VO_2_ in non-CF BQ described in the literature to identify patients who do or do not respond to an intervention, it appears that our home-based programme had a positive influence on exercise capacity, improving the functional capacity for physical performance, reflected by the change in baseline VO_2_ levels. However, as this is unprecedented, it is difficult to indicate the cut-off points to assess a change that could be considered significant for the “responder”. There is a series of cases that tried to explain the differences in the training capacity of young sedentary patients, placing the mean gain of VO_2_ max around 25% of the reference values, but with a range that went from no gain to the doubling of VO_2_ max (without factors such as age, sex or physical constitution being involved) [32]. Depending on the cohort, a statistically significant change may not be clinically important; the clinically significant change following intervention might be quite different. In cardiovascular diseases these values vary from 1 mL/min/kg to 3.6 mL/min/kg VO_2_ mL/min/kg [33,34], but even in respiratory diseases there are no definite differences.

On the other hand, responders appear to have microvasculature effects that would translate to a significant increase in SpO_2_ [31]. Current evidence suggests that there is a multitude of factors that influence the response to regular exercise training (e.g., training intensity, body composition, blood pressure, and cardiac function, among others) [34]. 

It is known that exercise training also improves other cardiovascular and metabolic components that may not be linked to significant changes in VO_2_ max but greatly contribute to an individual’s health status [34]. Changes in VO_2_ can be challenging as these changes are not typically seen in existing specific pulmonary rehabilitation programmes. However, it is known that any type of intervention should produce a substantial change in the patient’s situation, even general advice on exercise, as evidenced by the improvement in the parameters recorded in our CG. However, it seems clear that when the training has a minimum of supervision, these changes are more relevant. This means that the earlier a training programme is anticipated, the greater the changes will be. Instead, these studies show us the changes that occur with an intervention but do not identify which patients will or will not respond. 

On the other hand, although there were no changes in the other parameters, studies such as the one by Bar-Yoseph [4] in non-CF BQ support our data. They evaluated young patients (under 18 years old) using CPET to compare the CF, non-CF BQ, and control groups. It was observed that peak VO_2_ slightly decreased compared to the control group, with the ventilatory equivalent for oxygen (VE/VO_2_) and ventilatory equivalent for CO_2_ (VE/VCO_2_) within normal limits, with no differences found in the ventilatory characteristics of the groups. These ventilatory alterations appear to be present from a very early age, and the peak VO_2_ varied depending on the severity of the disease [35]. 

Although in this study, the mean age of the population differed from that of our patients, the findings in our study with respect to ventilatory alterations were similar. In contrast, in non-CF BQ, the worsening of the ventilatory parameters may not be related to the type of clinical or functional involvement or aetiology but rather to physical training and conditioning, and, because of this, the RER of <1.1 in this study can be explained. In addition, the correlation of FEV1 with maximal exercise power has even been noted in CF patients who have moderate–severe functional impairment [36,37]. Other studies indicate that low VO_2_ can be partly explained by the presence of low FEV1, as is the case with COPD [38]. Instead, in our patients, there was a mild–moderate functional impairment without a relationship with VO_2_ in those severe cases. 

Regarding the results of the 6MWT, in spite of all the benefits shown by this HPRP, there was a notable improvement in the walked distance in both groups, which was higher in IG [10,12]. However, if we compare our data with those of other studies, it becomes clear that the data differ widely from one study to another and may be influenced by the group of patients, mean age, ethnicity, BMI, and even the learning effect [26]. In summary, these data indicate that most of the variation in 6MWD can be attributed to between-patient variation [39]. In our case, an improvement of 50 m in the IG and 35 m in the CG was observed; although the change in the CG was not statistically significant, other studies showed an increase in the walked distance in the control groups studied [10,14,40]. However, the MICD in the IG exceeded twice the stipulated minimum difference [26] in relation to the CG. In this regard, in COPD (in a series of 519 patients), a high percentage of responders, when exceeding the MCID of 25 m and 6MWT, was identified as the best parameter to assess the responders and non-responders. Furthermore, the observed results were not related to age or disease severity [41]. In contrast to these, further studies are needed for non-CF BQ to establish an agreed cut-off point.

In relation to PA, in our research, we have observed that patients with non-CF BQ dedicate little time to moderate-high intensity physical activity; not to mention the fact of low energy expenditure, i.e., they are sedentary, as it has been confirmed in other studies in this field [42]. If we compare these results with a similar methodological design study (based on the HPRP with a duration of 8 weeks), in which PA was assessed with the international physical activity questionnaire, we can see that patients present with low PA that slightly improves the number of steps after the intervention, without reaching statistically significant differences [14]. It is worth noting that, although the participants did not show a relevant change in PA intensity, in the HPRP group, there was an increase in the number of daily steps, which was attributed to the training programme. In this sense, it is a challenge to modify the levels of PA in the patients since these are influenced by other factors that do not depend only on the training programme (social support, beliefs, climate…). Another study [43], with different assessment methods (using the accelerometer to assess PA and behaviour techniques based on the transtheoretical model), also presents similar results to ours. Although we did not use behaviour change techniques, our HPRP included educational workshops, positive feedback, and the self-management of symptoms, which were not officially collected but which we considered necessary for a more effective programme. However, the evidence on the role of health behaviour change in optimising and maintaining benefits in chronic respiratory diseases is disputed [44]. This supports the need to seek newer strategies or methods that motivate the patient to promote their physical activity and reduce their inactive time.

Based on the recommendations of the respiratory rehabilitation programmes, we designed a programme that is accessible to all levels, which required little material to carry out and is easy to understand for the participants. During its development, no incidents related to the training programme were reported, and although our adherence rate was lower than reported by other studies of similar design, we were able to see a high percentage of patients who responded to the recommended programme. 

In addition, it should be noted that although the adherence rate was lower than expected, in was higher in the IG than it was in the CG and similar to that described for classic hospital programmes in other patients with chronic respiratory diseases such as COPD and in some studies carried out specifically in bronchiectasis and with small sample sizes [4,11].

Our intention was to develop a training programme that can be carried out with the participants’ everyday tools, thus helping to promote long-term maintenance of the habit. However, there are programmes that describe the use of elastic bands [15], gym memberships [12], or inspiratory muscle training [13] that, in the long term, do not offer greater benefits than those programmes that use elements that are easily accessible at home. We also found that 3 days of training per week is sufficient to bring about changes in the perception of dyspnoea and even in exercise capacity.

On the other hand, HPRP vary in methodology and sample sizes, with evidence that physical training programmes show an improvement in symptoms, especially dyspnoea. Although improvement in dyspnoea is variable according to published series and types of programmes, a study of an HPRP in non-CF BQ (N 19) showed a decrease in the degree of dyspnoea [14], as well as in the domain assessing dyspnoea in quality of life questionnaires in favour of physical training [7]. Our data are consistent with the published results [14,15]; although our sample size is small, it is sufficient to demonstrate that these changes are significant and important. In contrast, in a recent clinical trial of home aerobic training, there was no change in dyspnoea or in symptom items on quality of life questionnaires [15]. Nevertheless, Lee et al. [7] dispel the doubt about the benefits of physical training, and the results are overwhelmingly in its favour.

After all this, we must point out that based on the training programmes for other chronic diseases and adapted to BQ; we have addressed the usefulness of an accessible and low-cost programme for patients with bronchiectasis, which is a strength of our study.

Finally, certain limitations of our study must be noted here. On the one hand, the number of participants was relatively small in each arm; therefore, the results of the study should be interpreted with caution. However, the sample size was sufficient to demonstrate significant differences between the groups. In addition, the patients who agreed to participate had an average of good lung function and, consequently, could have better exercise capacity. However, the most relevant limitation was the way of measuring adherence to the rehabilitation programme. Although the patients were offered a self-administered daily PA diary, this is a subjective measurement. Perhaps, for future studies, for example, the use of a specific mobile application for PA should be assessed, where data such as distance travelled, time, energy expenditure, or caloric expenditure, among others, are obtained at the time of the completion of said activity and, with this, objectively quantify what has been performed. However, it should be noted that, despite a high percentage of the participants not filling in the diary, adherence was much higher than that of the patients who only received recommendations, as previously mentioned, and that hospital programmes have a considerable drop-out rate, which is one of the key problems to be solved in rehabilitation.

## 5. Conclusions

It is clear that our home-based rehabilitation programme seems to be safe and improves exercise capacity, which was inherently low in our patients with non-CF BQ, physical activity, and symptoms to a greater extent than only standardised recommendations, although this basic intervention also has a positive effect. Therefore, it should be considered a fundamental strategy in the treatment of these patients. Nevertheless, more research in this field to define what type of programme, duration, and intensity is more profitable, as well as strategies to improve compliance are needed.

## Figures and Tables

**Figure 1 ijerph-19-11039-f001:**
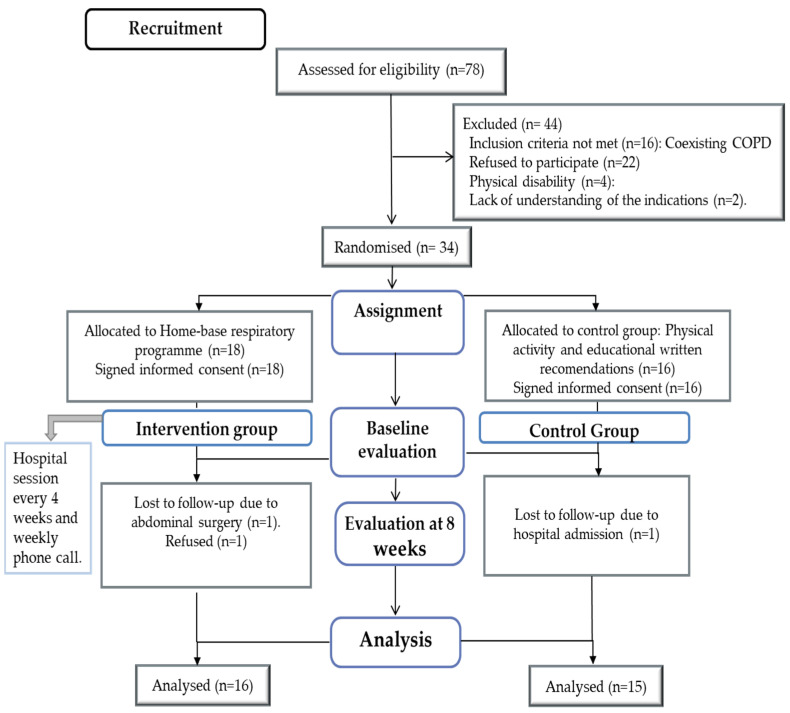
Flowchart of patients.

**Figure 2 ijerph-19-11039-f002:**
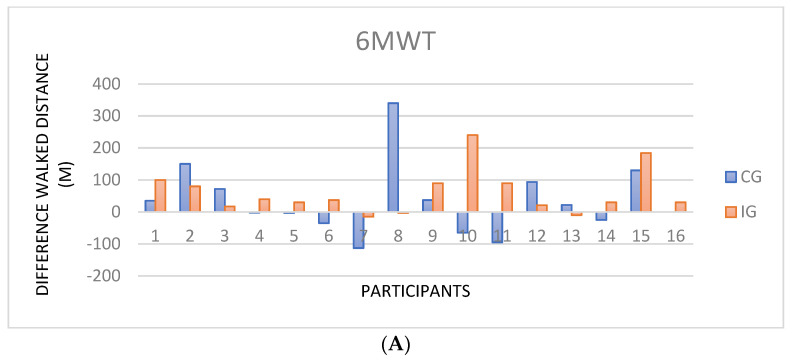
(**A**) The *y*-axis represents walked distance difference before and after HPRP, and the *x*-axis shows the subjects by groups. (**B**) The *y*-axis represents VO_2_/min/kg differences before and after HPRP, and the *x*-axis shows the subjects by groups.

**Table 1 ijerph-19-11039-t001:** Characteristics of the sample.

	Groups		*p*
	CG (*n* = 15)	IG (*n* = 16)	
Aetiology			
Post-infection	4 (26.66%)	6 (37.50%)	0.61 ^1^
Idiopathic	4 (26.66%)	1 (6.25%)	
Rheumatoid arthritis	1 (6.6%)	2 (12.50%)	
Immunodeficiencies	1 (6.6%)	1 (6.25%)	
PCD *	1 (6.6%)	1 (6.25%)	
Post-measles	1 (6.6%)	2 (12.50%)	
Sarcoidosis	1 (6.6%)	1 (6.25%)	
ABPA **	1 (6.6%)	1 (6.22%)	
Others	1 (6.6%)	1 (6.25%)	
General characteristics			
Women	10 (66.66%)	13 (81.25%)	0.30 ^1^
Age	59.43 ± 93.00	63 ± 6.14	0.33 ^2^
BMI	25.05 ± 2.52	26.35 ± 3.6	0.86 ^2^
Smoker	1 (6.66%)	2 (12.50%)	0.47 ^2^
Former smoker	2 (13.33%)	4 (25.00%)	0.29 ^2^
Never smoker	12 (80.00%)	10 (62.50%)	0.16 ^2^
PYI	2.70 ± 2.11	3.20 ± 1.83	0.45 ^2^
*P. Aeruginosa*	5 (33.33%)	6 (37.50%)	0.10 ^2^
Dyspnoea	2.07 ± 0.71	2.19 ± 0.63	0.28 ^2^
E-FACED	3.33 ± 1.73	3.44 ± 1.91	0.56 ^1^
Mild	7 (46.67%)	8 (50.00%)	0.54 ^1^
Moderate	7 (46.67%)	6 (37.50%)	
Severe	1 (6.66%)	2(12.50%)	
Lung function			
FEV1 L	1.84 ± 0.52	1.61 ± 0.44	0.56 ^1^
FEV1%	75.41 ± 28.44	71.88 ± 20.72	0.83 ^1^
FVC L	2.43 ± 0.71	2.17 ± 0.61	0.26 ^1^
FVC%	79.13 ± 24.14	79.13 ± 22.00	0.99 ^1^
FEV1/FVC	69.92 ± 14.73	74.33 ± 10.74	0.34 ^1^

Data are expressed as frequencies and percentages and as mean ± standard deviation. CG: Control Group, IG: Intervention group. * Primary ciliary dyskinesia (PCD), ** ABPA: Allergic bronchopulmonary aspergillosis. BMI: Body mass index. PYI: pack/year index. E-FACED acronym: forced expiratory volume in 1 s, FEV1%, chronic colonization by *P. aeruginosa,* radiological extension and dyspnoea, according to modified Medical Research Council (mMRC). In the *p*-value column, superscript ^1^ Student’s T test for independent samples and ^2^ Mann–Whitney U test; *p*-value before and after the HPRP in each group.

**Table 2 ijerph-19-11039-t002:** Dyspnoea, submaximal exercise, and physical activity.

	Control Group			Intervention Group		
	Before	After	*p*	Before	After	*p*
	Men ± SD	Mean ± SD		Mean ± SD	Mean ± SD	
6MWT						
Walking distance (m)	398.87 ± 67.10	433.13 ± 75.88	0.98 ^1^	401.2 ± 71.60	451.19 ± 67.99	0.006 ^1^
SpO_2_ (%)	95.4 ± 2.91	94.2 ± 3.24	0.42 ^1^	95.4 ± 2.93	95.8 ± 3.20	0.34 ^1^
Borg Scale	1.82 ± 2.3	2.33 ± 3.2	0.38 ^2^	2.45 ± 3.8	2.88 ± 1.31	0.30 ^2^
Physical activity						
Total steps per day (steps)	4.793 ± 3.236	4.824 ± 3.113	0.94 ^1^	4.578 ± 3.424	6.591 ± 3.482	0.007 ^1^
METS *	1.4 ± 0.3	1.26 ± 0.36	0.15 ^1^	1.42 ± 0.35	1.57 ± 0.35	0.47 ^1^
Lying time	9.03 ± 1.42	8.52 ± 1.02	0.01 ^1^	8.86 ± 1.44	7.91 ± 1.17	0.02 ^1^
Sleep time (hour)	6.79 ± 1.37	6.73 ± 0.9	0.03 ^1^	6.55 ± 1.06	6.85 ± 1.01	0.23 ^2^
Wearing time (hour)	22.34 ± 2.58	22.64 ± 1.85	0.75 ^1^	23.32 ± 0.28	22.85 ± 1.71	0.22 ^1^
Dyspnoea (mMRC)	2.07 ± 0.70	2.13 ± 0.64	0.06	2.19 ± 0.57	1.72 ± 0.05	0.04 ^1^

Data are presented as means and standard deviation (SD), separated into control group (CG) and intervention group (IG). * METS (metabolic equivalents needed to perform an activity). In the *p*-value column, the superscript ^1^ Student’s T test for related samples, and ^2^ Wilcoxon signed rank test for paired samples; *p*-value before and after the HPRP in each group.

**Table 3 ijerph-19-11039-t003:** Cardiopulmonary exercise test.

	Control Group			Intervention Group		
	Before	After	*p*	Before	After	*p*
	Mean ± SD	Mean ± SD		Mean ± SD	Mean ± SD	
W _peak_ (%)	62.7 ± 21.46	63.1 ± 17.34	0.242	67.2 ± 16.3	67.4 ± 11.9	0.92 ^1^
VO_2_ max (mL/min/kg)	13 ± 2.9	13.3 ± 3.4	0.57 ^1^	14.4 ± 3.2	14.9 ± 3.0	0.17 ^1^
VO_2_ peak (%)	59.8 ± 14.6	62.2 ± 14.14	0.30 ^1^	62.9 ± 15.8	66.8 ± 15.5	0.001 ^2^
HR (bpm)	117.13 ± 13.88	102.87 ± 29.04	0.041	116.25 ± 14.31	123.69 ± 20.5	0.051
HR(%)	73.6 ± 8.1	61.3 ± 26.9	0.22 ^2^	68.4 ± 10.1	61.7 ± 20.5	0.49 ^2^
O_2_/HR(%)	79.2 ± 12.2	81 ± 16.9	0.59 ^1^	83.4 ± 15.8	83.4 ± 15.8	0.51 ^1^
VE Max (L/min)	34.5 ± 7.2	33.5 ± 11.2	0.65 ^1^	39.8 ± 16.6	41.5 ± 18.4	0.46 ^1^
EqCO_2_	31.2 ± 3.0	31.5 ± 4.5	0.73 ^2^	32.4 ± 4.6	32.1 ± 4.1	0.37 ^2^
RER	0.99 ± 0.1	0.92 ± 0.1	0.09 ^1^	1.03 ± 0.01	1.01 ± 0.1	0.19 ^1^

Data are presented as means and standard deviation (SD), split into control and intervention groups. Variables: maximum work percentage (W peak%), peak oxygen consumption expressed as mil/min/kg and percentage respect of theoretical values (VO_2_ peak%), Heart rate (bpm) maximum heart rate percentage, refers to theoretical values, oxygen-pulse percentage (O_2_/HR%), maximal ventilation per minute (VE Max), respiratory exchange ratio (RER) and percentage breath rate (BR%). In the *p*-value column, superscript ^1^ Student’s T test for related samples, and ^2^ Wilcoxon signed rank test for paired samples; *p*-value before and after the HPRP in each group.

## Data Availability

Not applicable.

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
