# Peer review of "Exercise Capacity and Physical Activity in Non-Cystic Fibrosis Bronchiectasis after a Pulmonary Rehabilitation Home-Based Programme: A Randomised Controlled Trial"

_ijerph, 2022, doi:10.3390/ijerph191711039_

Round 1

Reviewer 1 Report (Previous Reviewer 2)

The authors have greatly improved their article compared to the initial version however it still persists several shortcomings and points to improve. The authors seem to confuse the x-axis and the y-axis in the figures, which is worrying! There are also a lot of small omissions to correct.

Methods

Line166: “was” calculated instead of “were”

Line 168:

HRmax Please correct the calculation formula that is not 210*(0.65*age) but 210-(0.65*age)

Line 177 : « Peripheral oxygenation » (SpO2)  do you mean Pulse haemoglobin oxygen saturation? Please correct.

“Initial and final heart rate” It’s quite imprecise . For initial do you mean at rest before exercise? For final do you mean during the last 30sec of exercise for example or immediately stopped.

Results

Line 211

About the severity of the disease and the distribution of patients in different grades. It seems to me that the total for every group should be 100% which is note the case here (46% + 67% + 41% while 154% for IG). In addition the syntax of the sentence is curious. Please check.

Line 232

Authors wrote that “50.2% participants of the IG confirmed that they had performed the recommended training program at least 3 times a week ”while it is written in the method (line 105 and line 119) that “it was  recommended to perform exercise at least five times a week !

Table 3

VE max (L/min) instead of L/m

Figures 2

To begin, this figure includes two figures which are very difficult to read and need to be reworked. After, there is a confusion on the axis labels, in figure A. I suppose that the abscissa show the subjects and that y-axis shows the change in VO2peak from baseline (mL/min/kg). In figure B x-axis is cut. It is also unreadable.

Line 279-280

Please standardize (comma or nothing) the presentation of the number of steps (instead of a point, a comma or nothing).

±33236. It’s a lot !

Figure 3:

« Título del eje ». Please delete

In the legend of the figure, the axis of abscissa and that of ordinate are confused and reversed.

The choice made to present the histogram series for one group,  followed without separation by the histogram series for the other group is not very relevant as already pointed out in the first reviewing.

Discussion

“Until now, researchers have had to extrapolate  data from the results of studies conducted in cystic fibrosis (CF)(19,20) or chronic  obstructive pulmonary disease (COPD)(24) to non-CF BQ because most authors have  used the 6MWT or the ISWT to evaluate exercise capacity during supervised  rehabilitation programs (29–31), instead of CPET, except in the case of home-based programs (32), which makes our study more relevant.” I don’t understand the meaning of this sentence and in particular the argument “because they have used 6MWT.…. instead of CPET”. Please rewrite more convincingly because the interest of the CPET is not really justified by your remarks.

“However, the changes we observed VO2 show”… Replace with:  “However, the VO2 changes we observed show…”

Line 317: “it was higher than in the control group….” add  “than in the IG…..”

Line 350 Change PHRP group by IG or change the sentence

Line 372 “We also found that training 3 days a week for 1 hour is sufficient to bring about changes…” This is appears in contradiction with your recommendation in the part method in which you asked subjects to perform physical activity “at least five times a week for 30 minutes”.

Line 383   “the mean response time to reach VO2peak …. improves ”. Not clear, what do you mean exactly? How do you call Wpeak the power at VO2 peak?

Author Response

Thank you.

Reviewer 2 Report (New Reviewer)

Dear authors, I am glad to review your manuscript. Despite being well-structured, some changes are needed.

Abstract

The first sentence from Background could be improved. Suggestion: The literature about patients with non-cystic fibrosis bronchiectasis is scarce, especially regarding the use of cardiopulmonary exercise tests to assess the effects of home-based pulmonary rehabilitation programs on these patients.

Page 1, lines 13 and 18

There seems to be more than one space between words when you mention “non-CF BQ”. I recommend the authors check the formatting. I noticed several formatting minor errors. Please, the authors should revise the formatting.

The second sentence should be placed prior to the first. It is reasonable to mention severe diseases and then the specific topic. I suggest the authors check these two sentences, mainly because of their similarities in both written and context.

Since the authors mentioned CPET in the objective, I recommend adding how PA was evaluated as well. Otherwise, I suggest just mentioning the interesting variables. Similarly, the methods lack information regarding the intervention and the evaluation protocol.

Page 1, line 23

(2.19±0.57 to 1.72±0.05 (p= 0.047)) replace by (2.19±0.57 to 1.72±0.05, p= 0.047)

I recommend doing the same for CG.

The results and conclusion include walked distance and physical activity, but this information was not presented in the methods. Please, consider revising.

The authors used the acronym for physical activity but did not include it in the conclusion.

Introduction

The second sentence needs a citation (page 1, lines 36-38). Where this information can be found?

The second paragraph must be improved (page 1, lines 39-42), especially the writing.

Although the third paragraph (page 2, lines 43-48) pointed out some relevant information, once again there is a lack of a proper citation. Please, I recommend revising.

I have the same concern about the fourth paragraph (page 2, lines 49-52).

I also suggest mentioning PRP at first and then PA. Both paragraphs could also be merged (lines 43-52).

When the authors mention changes in PA (page 2, line 55), I recommend using  "PA level" or "level of PA".

Although I agree that literature is limited on this topic, the authors have mentioned it in several parts of the Introduction section. Please revise in order to ensure a more concise, coherent, and objective introduction.

It is also worth mentioning that acronyms must be carefully checked. The acronym PRP, for instance, appears in line 57. However, the PRP was already mentioned in line 49.

Since the authors cited a study about changing PA levels on page 2 lines 75-76, I suggest revising the introduction and considering addressing PA from this paragraph.

Methods

Replacing gender by sex (page 3, line 102).

How were the participants recruited?

Considering the addition of some topics in this section, for instance, Study design and participants, Intervention or Pulmonary rehabilitation home-based programme. This could help the reader to understand the proposed methods. I also recommend mentioning ethics prior to the description of the intervention.

Since the evaluation was before and after the intervention, this information should be described before the intervention. Similarly, adding the measures as topics make the methods clearer to the reader, as well as allows a better understanding of the study design and protocol.

The authors did not mention any citations about the CPET. Please, revise this paragraph (page 4, lines 159-173).

On page 4 lines 176-178, add measurement units for each variable, as well as how was obtained this value.

Regarding PA assessment, please provide a reference for the classification. In addition, it is important to consider non-wear time. Did the authors evaluate sleep time? Why the use was from Wednesday to Sunday? Why were 5 days evaluated?

When describing the intervention, consider checking the description by using the Consensus on Exercise Reporting Template (CERT).

The sample size would be better placed in Statistical analysis.

I recommend the authors carefully revise the statistical analysis. The sentences are poorly written.

Results

Page 5, line 211

Lacking punctuation.

In the flowchart, the authors pointed out 44 excluded participants and other 16 that did not meet the criteria for participation, as well as other reasons. This information should be described in the text for providing clarification.

It is also unclear the duration of the intervention and if there was an actual follow-up. The authors should revise both the methods and results sections.

Page 5, lines 211-212

The sentence about the distribution needs improvement. In addition, the authors should adopt a pattern to report the findings since in this sentence you expressed the percentage as 46.67%, 33.3%, and 67%. Please, make the necessary changes. If this information is not reported in any of the tables, consider adding frequency and percentage.

Similarly, the authors should adopt a pattern in Table 1 as well. Some values seem incorrect, especially the frequencies and percentages. Please, double-check this table.

Regarding Table 2, I recommend merging it with Table 1 and dividing this information into the groups’ characterization.

The results need to be improved. A better description of the total sample, as well as the groups, should be added. For instance, it is important to mention that the sample was majority composed of elderly females and overweight participants.

Another table with results from dyspnea, and 6MWTD, among others.

There is no need to divide the results. I suggest removing topic 3.1. If the authors choose to maintain, I recommend adding other topics and dividing the section according to the assessments, for instance.

First of all, how do the authors explain the submaximal tests? Was it expected or not? Were the participants tolerant to exercise?

The authors mentioned changing the final HR, but I did not find this information in any table presented in the manuscript. Please, revise it.

How do the authors explain the HR change in both groups? Or this information was mistakenly reported (page 7, line 252)?

In the sentence on page 7 lines 252-254, please revise this information and rewrite the sentence. Similarly, the paragraph (page 7 lines 258-261) regarding PA results lacks clarity.

Although I understand the figures, using them instead of providing other tables with pre- and post-intervention evaluations leads to a confusing results section. I recommend the authors carefully revise this whole section.

Discussion

The first paragraph is composed of only one sentence. I recommend the authors revise and divide it into at least two sentences. In addition, provide a sentence mentioning the contribution of these findings to the literature and/or their practical implications.

COPD was already mentioned. Please, correct on page 9 line 301.

Although I understand the statement made by the authors regarding novelty, the main purpose of this study was not to determine the exercise capacity of non-CF BQ patients. The objective was to evaluate the effects of RPHP. Thus, I recommend the authors make this statement at the end of the discussion section as one of the main study strengths.

The third paragraph would be better placed near the end of the discussion, allowing the readers to understand the practical implications of this study. The authors also pointed out that no incidents were reported, but I am not sure whether this information is described in the results. Thus, I recommend the authors double-check and, if that is the case, consider adding clearer information about adherence and the absence of adverse effects during the intervention.

Since the main purpose was related to the exercise capacity, I strongly recommend that the discussion should start mentioning responders and findings regarding CPET and 6MWT.

When discussing exercise capacity, the authors suggested that the changes due to intervention could less clear for non-CF BQ in comparison to others. However, I partially agree with that. This result must be also linked to the intervention per se, which needs to be properly addressed.  

Regardless of the content, there are several errors in this section, including typo errors and unclear sentences. Therefore, the discussion needs to be entirely revised.

When addressing the low VO2, the authors are focusing on baseline data. Therefore, this should appear at the beginning of the discussion instead of on page 10. In addition, the authors need to discuss the effectiveness of the intervention, which was not properly made.

Including MCID enriches the discussion, I recommend doing the same for CPET findings (whether to justify the lack of positive results or to understand/propose possible explanations).

The authors repeatedly use “in general”. Please, revise it and maintain only when needed.

The authors mentioned the challenges of modifying a lifestyle. Besides that, I suggest the authors include a discussion about the evaluation of PA level and other aspects related to behavior change.     

Starting from line 360, the authors are discussing PA. In the next paragraph, there is a discussion about habits impacting dyspnea and exercise capacity and then, in the following paragraph, the text returns its focus on PA. This could be improved. Otherwise, it lacks focus and clear reasoning.

The authors cited Pehlivan and Bradley’s study. It would be interesting to understand methodological differences when comparing their study with the present study. The study used some type of behavior technique change or not, for instance. The measurement of PA was similar to that performed in the present study or not. What do we already know from behavior change in pulmonary diseases?

The findings must also be linked to the sample, which needs to be addressed.

In line 392, the authors stated that “Our data are consistent with published results”, but there is no study cited. Please, revise.

Despite mentioning the limitations, please provide some strengths from your study and also practical implications based on your findings.

Author Response

Thank you.

Round 2

Reviewer 2 Report (New Reviewer)

Although the manuscript needs to be double-checked in formatting and writing, it can be accepted in this version. It is also worth mentioning that the changes helped to improve the manuscript, as well as attended to the prior requests at least most of them. 

This manuscript is a resubmission of an earlier submission. The following is a list of the peer review reports and author responses from that submission.

Round 1

Reviewer 1 Report

An original article about home-based rehabilitation program in non-cystic fibrosis bronchiectasis was presented to my for review. In overall view work is interesting and preparing in good way. However, I have some comments and suggestions for Authors:

  1. In abstract authors forget to describe the control group, I think the CG shortcut should be explained. Additionally (however, this is a more technical note, the work should be re-edited and double spaces removed, e.g. line 24 and lack of spaces (line 52)).
  2. The introduction of the work is correctly described, introduces the reader to the subject. The aim of the study is correctly described too. However, I believe that it is worthwhile to describe the beneficial types of physical effort in more detail in the introduction to this paper, and to discuss what exercises are important. With what intensity the patient can/ should perform the efforts, how long and with what frequency. The practical approach to work is very important and to provide information to practitioners.
  3. I believe that information about the test and control groups should be included in the materialss and methods - e.g. discussing the age, sex, and number of respondents. 
  4. What maximum heart rate was set by the end of the test in Bruce's protocol. Why did the authors not use Bruce's Modified Protocol?
  5. I believe that the name of the tables should appear above it for better readability.

Reviewer 2 Report

This study deal with non-cystic fibrosis bronchiectasis patients. The goal of this study was to demonstrate the benefits of a home based physical rehabilitation program in these patients by comparison with only written advices and recommandations for a physical practice. In addition authors wanted to demonstrate the interest of the cardiopulmonary exercise testing as they condider it provides information on physical state and limitations that are more complete than that of submaximal exercise tests.

The major limit of this article relate to the writing or the presentation, in particular those of the method and the results which contain gaps or major blunders. Such weaknesses cast doubt on the quality of the study.

Material an Methods

It is not clear what was exactly adviced in the control group for physical activity

Line 77 you say that that the PRHP benefited for two hospital session but Fig 1 it appears that both groups had two hospital sessions. If true, what has been offered to the control group ?

Line 79 What was the reminder call ? Could you clarify a bit.

Line 81 Physical activity was controled with questionnaire. Do you think it is a reliable control ?

Measurement of variables

The different tests are not well enough describded as well as variables collected. For example the cardiopulmonary exercise test : duration of the stages and increase of the load at every stage? Pulmonary function : what maneuver performed. The 6MWT : a few words about its realisation and the measured variables should be added (distance and SpO2). The accelerometer :  when were performed the 3 days measurements before and after rehabilitation ? it’s too imprecise . Why 3 days and not a week ?. About E-FACED Scale : you detail the variables taken into account but they are not the same when listed under the table 3 !

Statistical analysis

A statistical paragraph is missing in the method, while below each table, the tests used are undeservedly repeated (for example below table 4). Please add a statistical part and then remove the description of the test performed below tables.

Throughout tables, when there is a column for statistical result, it should be written under the table what is the comparison performed (between groups or pre-post rehabilitation), also correct ±SD instead of ±DS (sometimes). Furthermore when p>0.05 commonly it is written NS, for non significant. Everywhere in the text write p=0.001 for example, instead of p 0.001.

Results

You must standardize in overall the article the way you speak of the 2 groups, for example Control group (CG) and interventional group (IG) but not for the latter, once study group , once PRHP group etc...

Lines 133-135 : This sentence is incomprehensible and should be rewritten.

Table 3

This table is not well presented and should be reorganized (5 columns (variable names , CG (pre and post) and IG (pre and post). Add variable units, for example 6MWT distance (m). PM6M : do you mean 6MWT ?

Table 4

This table is not well presented and should be reorganized (5 columns (variable names , CG (pre and post) and IG (pre and post). The title should be changed as those are variables at maximal exercise that are presented during CPET. It includes also too much variables that are not essential or redundant ((VO2max l/min ; O2/HR and O2/HR %, BR%) while maximal power (W) should be added. VO2max% : % of what ? If it refers to theoretical values add the bibliographic reference.

Lin 163-164 : If VO2 max criteria were not reached at maximal exercise (i.e. at least 2 of the 3 following criteria: (VO2 cap, maximal HR for age, RER≥1.1) we must not use VO2max but VO2 symptoms limited (VO2SL).

Lines 183-186 : authors say that there were none differences in exercise capacity variables as a function of the severity. Thereby the table 5 is not of interest. Just say it in the text.

Graph 1

Graph 1 should be untitled Figure 2.

This figure is not well presented and should also be reorganized. CG and IG « sticks » should appear nearby for every level for visual comparison. The ordinate axis is mysterious, what is the variable presented and unit ?

Table 6 does’nt appear needful.  Just includes major result it in the text.

Discussion

The discussion includes a succession of short paragraphs but is not really structured. It must therefore be revised and improved.

Lines 231-236 : this sentence is not correctly written and should be checked.

Line 243 : « with watts » watts is a unity of power probably do you want to say maximal exercise power.

Line 221 and line 255 : the increase in…  and in physical activity » probably do you mean the amount of physical activity, daily or weekly.

Lines 278 and 279 : please add the reference numbers

Conclusion

Quite short !!

Reviewer 3 Report

The authors of “Exercise capacity and physical activity in non-cystic fibrosis bronchiectasis after a pulmonary rehabilitation home-based program: a randomized controlled trial” have submitted a manuscript that presents a study with important methodological weaknesses to be considered.

Although the title and the intervention group were named as pulmonary rehabilitation, the intervention description includes strictly exercise instructions to be compared with physical activity indications as a control group. According to the most updated definition of pulmonary rehabilitation by the European Respiratory Society and the American Thoracic Society, such intervention it is not limited to exercise and should include at least education for self-management aiming long-term health behaviour change (Spruit MA, Singh SJ, Garvey C, et al. An official American Thoracic Society/European Respiratory Society statement: key concepts and advances in pulmonary rehabilitation [published correction appears in Am J Respir Crit Care Med. 2014 Jun 15;189(12):1570]. Am J Respir Crit Care Med. 2013;188(8):e13-e64. doi:10.1164/rccm.201309-1634ST). Because of this, this study cannot be presented as a pulmonary rehabilitation intervention. Furthermore, there is an inaccurate aspect to highlight, which is the fact that the authors describe the resistance training as walking or cycling, when such activities are examples of aerobic training (whereas resistance training is in accordance with what the authors described as strength training).

Also, one of the aims of the study as reported was to evaluate the impact on maximum exercise capacity by means of a cardiopulmonary exercise testing pre and post intervention. However, such methodology is not considered appropriate, as the maximum exercise capacity is a baseline assessment for exercise training intensity to be prescribed. Moreover, the authors describe an incremental maximal exercise test with the Bruce protocol, which is valid for a treadmill equipment and not for a bicycle ergo-spirometer as it is mentioned. Instead of an incremental exercise test, a constant-load exercise test would be appropriate for the purpose of measuring the intervention impact (Puente-Maestu L, Palange P, Casaburi R, et al. Use of exercise testing in the evaluation of interventional efficacy: an official ERS statement. Eur Respir J. 2016;47(2):429-460. doi:10.1183/13993003.00745-2015).

Also, a critical aspect not to disregard is the fact that the authors have not clearly identified the study main outcome and additionally sample size calculation is missing. Because of this, there is an enormous uncertainty about the statistical power of the study, and if the results presented are significant to provide any evidence or conclusions to be considered.

At last, the authors presented an introduction with reference to a systematic review (Bar-Yoseph R, Ilivitzki A, Cooper DM, et al. Exercise capacity in patients with cystic fibrosis vs. non-cystic fibrosis bronchiectasis. PLoS One. 2019;14(6):e0217491. Published 2019 Jun 13. doi:10.1371/journal.pone.0217491), which is in fact a cross-sectional study.

Taking in account all the above-mentioned aspects, the reviewer’s overall recommendation is to reject this manuscript publication.